# TextIT: Inference-Time Representation Alignment for Improved Visual Text Generation in Diffusion Models

**Abhikhya Tripathy**
Adobe Research, India
abhikhyat@adobe.com

**Aishwarya Agarwal**
Adobe Research, India
aishagar@adobe.com

**Srikrishna Karanam**
Adobe Research, India
skaranam@adobe.com

**Balaji Vasan Srinivasan**
Adobe Research, India
balsrini@adobe.com

## Abstract

Recent advances in text-to-image diffusion models have shown remarkable performance in generating realistic images from text descriptions. However, high-quality visual text generation in generated images remains a major challenge. Gibberish text generation is particularly problematic when the model has to generate proper nouns and text that is not commonly present in training data. Unlike existing methods to improve visual text generation which are based on data-intensive and time-consuming fine-tuning approaches, we propose an inference-time representation alignment algorithm, **TextIT**, that does not need additional data or training. First, we propose an inference-time self-attention manipulation loss that exposes and aligns latent intermediate self-attention (SA) representations governing visual text generation with those of correctly-rendered text. Next, we impose fine-grained control over the generated text by aligning character-wise control points, obtained through self-attention map vectorization, with ground truth character control points. We provide evidence that inference-time representational manipulation enables controllable and interpretable improvements in text-to-image generation, validating our method with character and word-level visual text generation results that retain the overall generative diversity of diffusion models.

## 1 Introduction

In recent times, diffusion-based generative models (Balaji et al. [2023], Gu et al. [2022], Ho et al. [2020], Rombach et al. [2022], Saharia et al. [2022], Song et al. [2022], Zhao et al. [2023]) have become very popular for various text-to-image generation tasks due to their remarkable ability to generate high-quality, realistic and diverse images, successfully outperforming earlier methods (Dhariwal and Nichol [2021]) based on Generative Adversarial Networks (GANs) (Goodfellow et al. [2014], Radford et al. [2016]) and Variational Autoencoders (VAEs) (Kingma and Welling [2022], Rolfe [2017]). However, despite their impressive performance in producing high-quality images, one limitation with most existing approaches is their inability to render accurate text in the generated images (Daras and Dimakis [2022]). The images often contain text that is ill-formed, illegible and inaccurate. It is frequently observed that the rendered text contains strokes and symbols that do not match any glyphs in the given language. Consequently, the overall quality of images that require visual text decreases, making diffusion models a bad fit for such image generation tasks. Such tasks include generation of logos, book covers, newspapers etc.

The rendering quality is especially bad for text that is unlikely to be present in the image-text paired training data of these models. Examples include proper nouns—rare names of people, places and

objects, and words that are not commonly used in the language. The inability of current diffusion models to accurately render such text can be largely attributed to the lack of sufficient image-text data which contains annotations for the text content in the images. Thus, during training, these models fail to acquire robust internal representations for text, unlike the stronger representations they learn for objects and scenes. Furthermore, during inference, there is no loss function that explicitly provides guidance for the rendered text or enforces its alignment with the generation prompt. These factors, along with the fine-grained complexity of visual text glyphs, make this a difficult task.

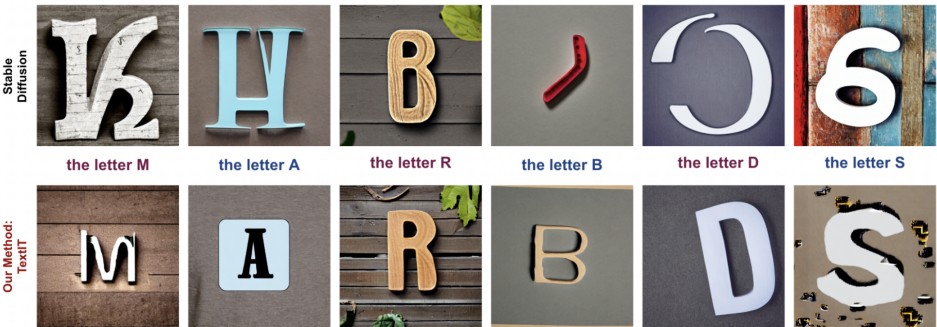

Figure 1: To tackle the inability of current text-to-image diffusion models in Visual Text Generation, we propose **TextIT**, an inference-time optimisation approach that exposes and aligns intermediate self-attention representations. The above results show that our method enables significant improvements in text rendering compared to the baseline Stable Diffusion, while preserving overall image characteristics such as background, color palette, and style.

To address the challenge of visual text generation, recent works have explored different directions. Some techniques have focused on replacing CLIP with large language models such as T5 (Balaji et al. [2023], Saharia et al. [2022]) as the text encoder in diffusion models, showing improved text rendering. Others showed that the text encoders used in standard text-to-image diffusion models lack character-level spelling information because they use Byte-Pair Encoding (BPE) tokenizers. They have shown that replacing this with character-aware models such as ByT5 can improve results (Liu et al. [2023]). Instead of using a subword vocabulary like most other pretrained language models (BERT, XLM-R, T5, GPT-3), the ByT5 model operates directly on UTF-8 bytes (Xue et al. [2022]). Additional approaches provide explicit text positional or content guidance, such as segmentation masks and glyph images in GlyphDraw Ma et al. [2023], GlyphControl Yang et al. [2023], and TextDiffuser Chen et al. [2023a], and fine-tuning a large language model for position and layout planning for TextDiffuser-2 Chen et al. [2023b]. AnyText Tuo et al. [2023] encodes glyphs, positions and masked images into the latent space and uses stroke information as additional embeddings along with image-caption embeddings, while Diff-Text Zhang et al. [2023] proposes a training-free approach that conditions diffusion models on Canny-detected edge images of rendered target text, combined with localised attention constraints to appropriately position the scene text. While these works highlight the value of controlling representations for text, they often depend on fine-tuning which creates the necessity for appropriate data and computational requirements, or rigid conditioning such as Diff-Text (Zhang et al. [2023]) which incorporates text edge image and local guidance into the UNet itself limiting flexibility and creative diversity of generated outputs. Furthermore, these methods often fail to accurately render text that is unseen during training such as rare words, proper nouns, and text borrowed from a different language.

To overcome these limitations without requiring additional training or fine-tuning, and without altering conditional inputs to the base diffusion model, we introduce **TextIT**, an inference-time optimisation algorithm to improve visual text generation in text-to-image diffusion models. Our method dynamically adjusts intermediate self-attention maps during inference so that they better align with reference maps of correctly-rendered text. These reference maps are obtained using the Skia graphics library by automatically rendering the target text in a standard font and style, and we further vectorize them to extract Bézier curve control points. Such control points provide a novel, interpretable handle on text glyph geometry, offering a representation-level view of how text is encoded. By minimising a loss between these control points and those extracted from self-attention maps of correctly-rendered text, TextIT exposes and aligns the internal representations that drive text generation. The control points provide flexible parametric controllability over the text generation

ability. Importantly, this is achieved without modifying model weights, providing evidence that inference-time representational adjustments alone can significantly enhance generative behavior. Experimental results demonstrate the efficacy of our method in generating visual text with greater accuracy and controllability.

To summarize, our key contributions in this work are:

1. A completely training-free, inference-time-only approach that improves text rendering in diffusion models by aligning internal self-attention representations. Our work is the first representation-level inference-time intervention for accurate visual text generation.

2. Our method handles the problem of unseen text data, as ground truth attention maps can be automatically obtained for any text (given we know the text to be rendered from the generation prompt) using a rendering library, such as Skia [1].

3. We provide evidence that inference-time representational manipulation not only improves text fidelity, shown through character and word-level generations, but also acts as a probe into how diffusion models encode structured glyphs, suggesting future directions for controllability and interpretability.

## 2 Related Work

### 2.1 Text-to-Image Diffusion Models

Denoising Diffusion Probabilistic Models (DDPMs) have had disruptive effects in the field of image generation from text prompts, having recently replaced methods based on GANs and VAEs as the most popular choice for the task (Daras and Dimakis [2022], Ho et al. [2020], Song et al. [2022], Dhariwal and Nichol [2021], Saharia et al. [2022], Ramesh et al. [2022], Rombach et al. [2022]). Imagen Saharia et al. [2022] introduces standard large language models (T5-XXL text encoder) into the image generation pipeline, in place of the CLIP text encoder, and demonstrates comparable or superior overall image quality, while improving performance in visual text rendering. On the other hand, eDiff-I Balaji et al. [2023] concatenates the CLIP text embeddings with the T5 text embeddings to leverage the capabilities of both text encoders. In addition to these aforementioned pixel-level diffusion models, Stable Diffusion Rombach et al. [2022] is a Latent Diffusion model that performs the diffusion process in a downsampled latent space, and uses a powerful CLIP text encoder to provide conditional generation information in text-to-image synthesis. Stable Diffusion-XL (SDXL) (Podell et al. [2023]) improves on top of Stable Diffusion by using a 3 times larger UNet, a second text encoder and another refinement model to enhance image generation quality through image-to-image techniques. In our work, we adopt Stable Diffusion as our base model.

### 2.2 Visual Text Rendering

Despite the success of diffusion models in text-to-image synthesis, accurate visual text rendering remains a persistent challenge. A number of works address this using additional conditioning or fine-tuning with new data. For example, GlyphDraw Ma et al. [2023] and GlyphControl Yang et al. [2023] introduce glyph-based inputs, with the latter also incorporating multi-line layouts and positional information. TextDiffuser and TextDiffuser-2 (Chen et al. [2023a,b]) provide character-level segmentation masks and layout planning through large language models, while AnyText Tuo et al. [2023] encodes glyphs, positions, and strokes into the latent space to enhance rendering accuracy. Other approaches include Diff-Text Zhang et al. [2023], which uses edge maps with local attention constraints, and DreamText Wang et al. [2024], which reconstructs the training process with refined character-level guidance. Another line of work focuses on improving the text encoders within diffusion models. Imagen Saharia et al. [2022], eDiff-I Balaji et al. [2023], and DeepFloyd IF leverage large-scale language models such as T5 to strengthen text representations. At the same time, methods like ByT5 Xue et al. [2022] and Glyph-ByT5 Liu et al. [2024b] replace or fine-tune character-blind encoders to capture glyph-level structure more effectively. In contrast, our work is markedly different from these by being completely training-free, avoiding the overhead of fine-tuning data, training time and resources, and instead operating by directly manipulating self-attention representations at inference time to improve alignment between prompts and rendered text.

---

[1] https://skia.org/docs/user/api/

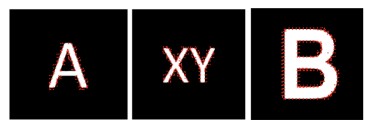

Figure 2: Control Points generated from Bezier curves for Skia-rendered ground truth SA maps

# 3 Proposed Approach

## 3.1 Latent Diffusion Models and Self-Attention

We start with a brief review of diffusion models and the associated self-attention computation mechanism. We adopt Stable Diffusion (SD) (Rombach et al. [2022]) as our backbone. The architecture of Stable Diffusion uses a Variational Auto-encoder (VAE) which, given an input RGB image $x$ of dimensions $H \times W \times 3$, uses an encoder $\mathcal{E}_v$ to transform it into a latent representation $z$ of dimensions $h \times w \times c$, where $\alpha = \frac{H}{h} = \frac{W}{w}$ is the downsampling factor of the VAE and $c$ is the latent feature dimension. After the diffusion process, decoder $\mathcal{D}$ is used to reconstruct the image $x$ from the latent space representation $z$.

The diffusion process is carried out in this low-dimensional latent space, where a conditional UNet (Ronneberger et al. [2015]) is used as a denoiser $\epsilon_\theta(z_t, t, y)$ to predict the noise in the noisy latent $z_t$. Layers of the UNet consist of a residual block, a self-attention block and a cross-attention block. Here, $t$ is the current time step and $y$ is the generation condition information which is generally text (in text-to-image models) that is projected to an intermediate representation $\tau_\theta(y)$ by a text encoder $\tau$ (such as CLIP Radford et al. [2021], T5 Raffel et al. [2023], ByT5 Xue et al. [2022]). However, $y$ can also belong to other modalities, e.g., segmentation maps, canny edge maps etc. This encoded condition information $\tau_\theta(y)$ is then mapped to the intermediate layers of the UNet via a cross-attention mechanism between projections of both image and text modalities. Moreover, the residual block of the UNet convolves image features $\phi_t^{l-1}$ from the previous layer to produce intermediate feature maps $f_t^l$ for a given layer $l$. Self-attention is computed between projections of these feature maps into queries, keys and values. This operation gives rise to a set of self-attention maps $\mathbf{A_t} \in \mathbb{R}^{(r \times r) \times (r \times r)}$ , where $r = 8, 16, 32, 64$, which we extract and use. The overall training objective is to minimise the following loss:

$$\mathcal{L}_{\mathcal{SD}} = \mathbb{E}_{\mathcal{E}(x), y, \epsilon \sim \mathcal{N}(0,1), t} \left[ \| \epsilon - \epsilon_\theta(z_t, t, \tau_\theta(y)) \|_2^2 \right] \tag{1}$$

In inference, a latent code $\mathbf{z}_T \sim \mathcal{N}(0, 1)$ is sampled and the prompt $y$ is encoded into $\tau_\theta(y)$. Then, $T$ denoising steps are run using the UNet denoiser $\epsilon_\theta$ to obtain $\mathbf{z}_0$, which is decoded using $\mathcal{D}$ to obtain the generated image $x' = \mathcal{D}(\mathbf{z}_0)$.

## 3.2 Control Points

As mentioned in Iluz et al. [2023], Thamizharasan et al. [2024], modern typeface formats, such as TrueType and PostScript represent the glyphs contained in the alphabet using a vectorized graphic representation of their outlines. The individual character glyphs are made up of outline contours that are represented by a collection of lines and Bézier/B-spline curves. These Bézier curves are defined with a set of control points which can be extracted from the vector representation of the corresponding image. An example is illustrated in Figure 2, where we can see the constituent control points around the glyph of the character 'B'. These control points have been automatically generated using our pipeline's vectorisation module, Mang2Vec (Su et al. [2023]), an approach for vectorising raster black-and-white mangas, using a trained Deep Reinforcement Learning model. We use this work for extracting control points because our intermediate and ground truth self-attention maps are gray-scale images. The Bézier curve C is represented in the following form:

$$Q : \; x1 \; y1, \; x \; y$$

The second set of coordinates $(x, y)$ specifies where the curve should end. The set $(x1, y1)$ is the control point of the curve. The control point essentially determines the slope of the curve at both its start and end points. The Bézier function then creates a smooth curve that transfers from the slope established at the beginning of the line, to the slope at the other end.

From Fig. 2, we see that with extracting control points, we can obtain point clouds along the outlines of the glyphs of the visual text. Thus, we use these as an interpretable representation of glyph geometry, and extract and optimise point clouds generated around the glyphs in our self-attention maps. This enables fine-grained control of shape and structural details of the generated visual text.

## 3.3 Proposed Approach

As mentioned earlier, prior works have tackled the issue of visual text rendering via different text encoding strategies or using additional input conditions, and fine-tuning on appropriate datasets. This makes these methods data-intensive, computationally demanding and difficult to generalize to out-of-domain scenarios. To this end, our approach comprises of a training-free inference-time optimisation, where we dynamically manipulate self-attention maps of the generated images during denoising steps, to align them with correctly-rendered ground truth self-attention maps.

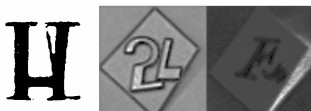

Figure 3: SA maps from intermediate denoising steps

Self-attention maps in diffusion models provide an interpretable, semantically-meaningful intermediate representation for steering visual content by encoding long-range spatial dependencies and revealing task-relevant image structures (Chefer et al. [2021], Carion et al. [2020], Mehrani and Tsotsos [2023]). Chefer et al. [2023] have shown that targeted manipulation of the attention mechanism can be used to improve semantic fidelity to input prompt during inference. Liu et al. [2024a] demonstrate that self-attention maps capture the geometric and shape details of the generated image, and modify those maps during denoising to achieve controllable image edits. Inspired by these works, which validate attention as an actionable control mechanism for generative steering, we manipulate self-attention maps in order to exploit the correspondence between these maps and the glyphs of generated text renderings. We demonstrate this correspondence in Fig. 3, which shows intermediate self-attention maps of some generated visual text, where the glyph structure of the text to be rendered is clearly visible. The self-attention map which is used for visualization, and for our inference-time optimisation, is actually obtained by retaining the first principal component after performing Principal Component Analysis (PCA) on the self-attention maps extracted directly from the UNet (we provide exact details on how we compute and visualize the intermediate self-attention maps in the Supplementary Material). This emergence of the final glyph structure further validates utilizing intermediate self-attention maps to detect and rectify incorrect text glyphs, as the self-attention maps at any given denoising step drive the final generation. Therefore, manipulating these intermediate maps to match the target text shall allow controlling and steering the quality of generated text towards the one specified by a user in the input prompt.

The overall architecture of our method, **TextIT**, is visually represented in Fig. 4. We first extract self-attention maps of the image being generated from the intermediate denoising steps of the diffusion process, and then perform self-attention manipulation by proposing two different losses computed between these attention maps and the ground truth attention maps.

We take into consideration those cases of generation where the users want the image to contain specific text. The ground truth attention maps are obtained from using a standard 2D graphics library, such as Skia, to render the target text that is to be present in the image, in a fixed standard font and in black-and-white (Refer to Fig. 2 for examples). The style and size in which the ground truth is rendered remains fixed for all generations. The target text can be determined from the text prompt passed into the text-to-image model, either explicitly present in the prompt, or can be implicitly inferred from it using a large language model, such as GPT-3.5.

We next elaborate the proposed losses computed between the intermediate self-attention maps and these ground truth maps in order to manipulate the self-attention maps during denoising:

1. **Self-Attention Loss, $\mathcal{L}_{SA}$:** We compute a direct pixel-intensity loss between the images of the self-attention maps and the ground truth maps. Both the self-attention and the ground truth maps are single-channel gray-scale images. For computing the loss, we use the Focal

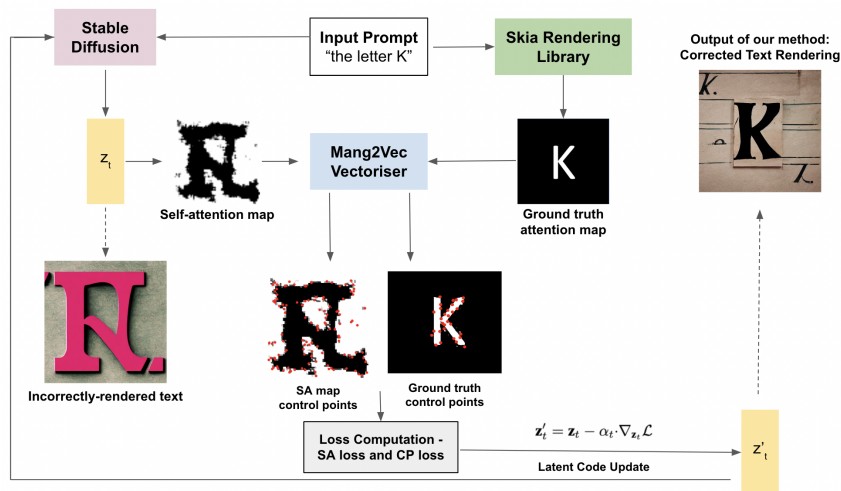

Figure 4: Overall Architecture of our proposed method, **TextIT**

Loss (Lin et al. [2018]), which is used abundantly for object detection tasks. We calculate this loss between the SA maps and the ground truth maps. Observations from our experiments have shown that this loss function leads to better performance compared to other losses, such as MSE Loss. We provide more details about Focal Loss in the Supplementary Material.

2. **Control Points Loss, $\mathcal{L}_{CP}$:** After obtaining the self-attention map for the particular intermediate inference step, we pass it into our vectorising module, Mang2Vec, which takes an input grayscale image and outputs its vectorised SVG form. From the outputs of this pipeline, we can extract the control points that constitute the vectorised form of our images from the Bézier curves. We extract control points from both the intermediate self-attention maps and the black-and-white ground truth maps. Some examples of control points extracted for ground truth maps are shown in Fig. 2. We then compute the loss between these two sets of 2-Dimensional points using MSE.

Once we have computed these losses, we use a combination of the two as our overall loss $\mathcal{L}$ for the inference-time optimisation routine. The overall loss is defined as: $\mathcal{L} = \beta_1 \mathcal{L}_{SA} + \beta_2 \mathcal{L}_{CP}$,

where $\beta_1$ and $\beta_2$ are hyper-parameters that control the time-steps for which the Self-Attention loss and the Control Points loss are to be applied respectively, as part of the overall loss. For the time-steps for which we want to apply only CP loss, $\beta_1 = 0$ and $\beta_2 = 1$, and for the time-steps where we want to apply only SA loss, $\beta_2 = 0$ and $\beta_1 = 1$. In case we want to apply both losses, $\beta_1 = \beta_2 = 1$, and in case of neither, $\beta_1 = \beta_2 = 0$.

In each of these optimisation paradigms, once we compute the overall loss $\mathcal{L}$ for a particular denoising timestep, we need to update the latent code at that timestep $\mathbf{z}_t$ such that it changes in the direction of minimising the overall loss. This is based on our intuition that, given the definitions of the losses used, minimising these would lead to more accurate text rendering. To achieve this, we perform the following latent update:

$$\mathbf{z}'_t = \mathbf{z}_t - \alpha_t \cdot \nabla_{\mathbf{z}_t} \mathcal{L} \tag{2}$$

where $\alpha_t$ is the step size of the gradient update, and $\nabla_{z_t}$ is the gradient of the loss computed with respect to the image's latent code at denoising timestep $t$. The updated $\mathbf{z}'_t$ is now used in the next denoising timestep in place of $\mathbf{z}_t$, and this entire gradient update step is performed for all the timesteps that we perform inference-time optimisation for.

## 4 Experiments

We perform experiments with our proposed method in order to show Proof of Concept results for our method, by implementing our optimisation strategy on top of our baseline diffusion model and showing improved results for visual text rendering of single characters in the English alphabet. Our

text prompts into the text-to-image models thus take the form of "The letter A", "The letter B" and so on. We observe that our method significantly improves rendering of single characters in English when compared to our baseline model, in a completely training-free approach.

**Implementation Details:** We use Stable Diffusion 1.4 as the backbone for our proposed inference-time optimisation. However, our code is compatible with all versions of Stable Diffusion models. As described earlier, our contributions include two losses computed from the latent code representation that are optimised during inference time. These are the Self-Attention loss, $\mathcal{L}_{SA}$, and the Control Points loss, $\mathcal{L}_{CP}$. The $\mathcal{L}_{SA}$ loss computes the difference in pixel-intensity between the ground truth and intermediate self-attention maps. We found that the Focal Loss is best-suited for this task.

On the other hand, the $\mathcal{L}_{CP}$ loss computes the alignment between the control point clouds extracted from the ground truth and intermediate self-attention maps. To align these control point clouds, we use the MSE loss. The experimental setup for our inference-time optimisation routine is as follows:

- Self-Attention loss $\mathcal{L}_{SA}$ with Control Points loss $\mathcal{L}_{CP}$:

  Here, we use the Focal loss function to compute the SA loss between the ground truth and intermediate maps and align them in pixel-intensity space. Further, to align the control point clouds extracted from the respective attention maps, we use the MSE loss function. In this setup, we apply only the SA loss for the first 5 timesteps and only the CP loss for the next 20 timesteps. Thus, the overall loss function looks like ($t$ is the inference time-step number):

$$\begin{aligned} \mathcal{L} &= \mathcal{L}_{SA}, \ \text{ for } t \leq 5 \\ \mathcal{L} &= \mathcal{L}_{CP}, \ \text{ for } 5 < t \leq 25 \end{aligned}$$

For all our experiments, the total number of diffusion denoising steps is 50. Of these, we perform inference-time optimisation for only 25 denoising steps. The choice to optimise only during the first 25 steps is based on both empirical analyses and prior work on attention-based steering (**Attend-and-Excite** Chefer et al. [2023]). We have leveraged studies that suggest that diffusion attention mechanism follows a coarse-to-fine progression, early step representations capturing high-level spatial structures and middle steps capturing finer geometric details (Yue et al. [2024], Park et al. [2023]). Thus, we apply the pixel-based SA loss in the early denoising steps to optimise spatial and semantic structures, and the CP loss in the middle steps for fine-grained text glyph alignment. We observe that applying SA loss beyond the very early steps has little effect on the intermediate latents, thus we limit it to the first 5 steps. We do not apply CP loss beyond 25 steps to avoid compromising the generation diversity and creativity of the diffusion model, to ensure the latents remain in-distribution with respect to the pretrained model, and to decrease the total time and resources needed for inference.

**Baselines:** We compare **TextIT** with four different text-to-image diffusion models on the task of single-character visual text rendering. The first model we compare with is the Stable Diffusion model, which is the baseline that we implement and test our method on top of. We also compare our method with recent works focused on improving the text rendering ability of diffusion models, namely, TextDiffuser, TextDiffuser-2 and AnyText. One thing to note here is that the paradigm of operation of these three models is very different from **TextIT**. While these models use various additional conditioning information and have been fine-tuned on appropriate data, our method is completely training-free. In the Supplementary, we compare separately with Diff-Text because it is closest to our paradigm. We omit recent works such as Wang et al. [2024], Liu et al. [2024b], since these are also training-based methods, a category already well-represented among our baselines.

**Evaluation Metrics and Quantitative Results:** Quantitatively, we compare **TextIT** with four different text-to-image diffusion models on the task of text rendering with diverse generations. First, we compare with Stable Diffusion, which is the baseline that we implement and test our method on top of. We also compare with recent works focused on improving text rendering - TextDiffuser, TextDiffuser-2 and AnyText. The paradigm of operation of these models is very different from **TextIT**, making a quantitative comparison difficult. While these models use various additional conditioning information and have been fine-tuned on appropriate data, our method is completely training-free and inference time-only.

We use three quantitative metrics to compare our method with prior models: OCR accuracy, CLIP-Score, and FID score. For OCR accuracy, we employ PaddleOCR (Du et al. [2020]) on the task

| | Stable Diffusion | TextDiffuser | TextDiffuser2 | AnyText | TextIT |
|---|---|---|---|---|---|
| OCR | 33.3 % | 34 % | 60 % | 65 % | **68 %** |
| CLIP | 23.19 | 21.91 | 23.98 | 25.23 | **27.31** |
| FID | - | 320.1 | 361.5 | 259.7 | **220.6** |

Table 1: Quantitative Results

of single-character generation. A generated character is considered correct if at least one detected character matches the prompt, focusing on whether our method can render the intended glyph, and on representational alignment. We observe that models sometimes produce extra characters beyond what is specified in the prompt, which likely arises because models are trained on a lot of multi-character text. The generation of image artifacts which are not specified in the prompt is a common issue with the base Stable Diffusion model. We decide not to penalize this, since that isolates the effect of our algorithm from unrelated artifacts introduced by the base model. Future work, and extension to newer models, will incorporate stricter metrics and a focus on both text accuracy and artifact suppression. We evaluate on 30 prompts corresponding to randomly selected letters, covering all 26 English alphabet letters, with seeds fixed across models for consistency. Although this is a relatively small benchmark, it captures all 26 letters and adding more letters would serve as a multipler, leading to redundant averaging over the comparison for some letters. For CLIPScore and FID, we assess word-level generation to provide a more comprehensive comparison. CLIPScore is computed between prompts and images for 20 short word generations, while FID is computed between the outputs of Stable Diffusion and each baseline (including TextIT) for the same 20 prompts. Word generation is performed letter by letter, ensuring coverage of all characters in the alphabet, and we will expand this benchmark for the final version. We compare with Stable Diffusion outputs since all these works are built on top of it. These metrics serve not only as benchmarks for accuracy of text generation but also as indirect probes of how well self-attention representations align with target text structures. Table 1 clearly shows that our method performs the best in terms of all our metrics, showing the alignment of our outputs with text prompts as well as base SD outputs.

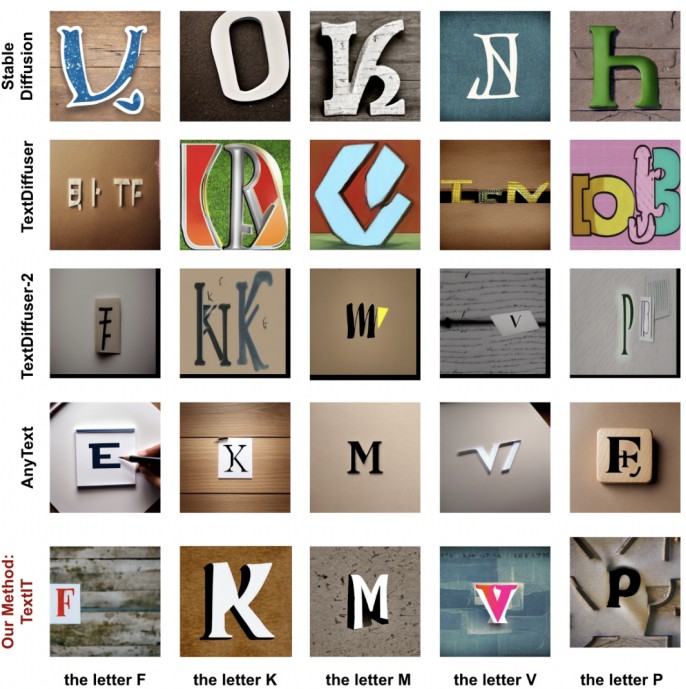

Figure 5: Comparison of **TextIT** with baselines for single-character visual text generation

**Qualitative Results:** Fig. 5 shows compared qualitative results of our method with our baselines, in the task of visual rendering of single-characters. As we can see, **TextIT** significantly outperforms the other methods in visual quality of generated single-character images, thus validating our Proof-of-Concept. An important point to note is that, apart from the glyph correctness and legibility, the

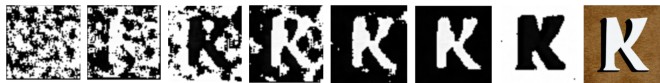

Figure 6: Evolution of SA maps through inference timesteps using **TextIT**, and the generated image.

style, colors, theme, and semantics of the rest of the generated image remain close to the base model generation. This demonstrates that our method and our choice of ground truth attention maps do not limit the font, style or diversity of outputs. We provide more qualitative results for single characters in the Supplementary. Note that other methods are bad at generating single characters even when the prompt explicitly asks for them.

In Fig. 6, we would also like to show how the self-attention maps evolve with denoising steps as our method progresses, along with the final fully-denoised output.

**The Need for CP Loss:**   Fig. 7 shows that combining CP loss with SA loss yields more accurate text, capturing fine-grained glyph shapes. SA loss is a pixel-based loss and, on its own, it aligns the pixel intensities and appearance of attention maps which does not provide detailed control, while CP loss aligns control points which are generated around the glyph edges, thus providing fine-grained geometric control. However, CP loss alone can be unstable as the computed loss could diverge since control points follow glyph shapes closely, so we combine it with SA loss for best results.

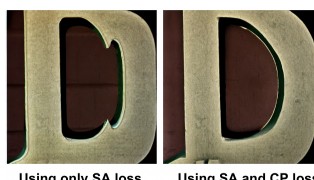

Using only SA loss    Using SA and CP loss

Figure 7: Qualitative ablation study. The use of Control Points loss in addition to Self-Attention loss provides fine-grained glyph shape refinement, leading to more accurate text rendering.

**Rendering Short Text:**   We present qualitative results for rendering short English text, which we take to be words with between 3 to 5 letters. This set of results provides the proof-of-concept for improved small words generation using **TextIT**. Specifically, we focus on the following 3 types of short words to show that our method can help render any arbitrary short text correctly, including uncommon words and proper nouns which have been a major pain point for diffusion models: non-proper nouns (both common and uncommon) which baseline Stable Diffusion is unable to render accurately, names of persons (including names not common in the English language), and names of places. Fig. 8 shows qualitative results for generating the above three categories of short words using **TextIT**, and their comparison with baseline SD generations for the same seeds and prompts.

To render short text, we first use an OCR model to detect bounding boxes in the SA maps containing characters. Each region is divided into $n$ segments, where $n$ is the number of target characters, and independent losses are computed against the ground-truth maps of corresponding characters for each segment. The latent is updated in a loop for each inference-time optimization step. By optimizing at the character level, our method can render arbitrary words, including proper nouns and unseen terms, since every word decomposes into characters. We use character level optimization so that our approach is robust and not tied to any specific dataset of words.

## 5   Conclusion & Future Work

In this paper, we introduce an inference-time optimization approach that manipulates and aligns self-attention representations in diffusion models, improving visual text generation without additional data or fine-tuning. Our results show that latent representation alignment makes rendered text more faithful to prompts while preserving overall diversity of generated images. This is the first representation-level inference-time intervention for visual text, and the control points formulation offers interpretable controls which could enable interactive human-in-the-loop editing. The same method can be extended to other structured elements such as tables, infographics and logos, highlighting the potential of inference-time representational control for robust and controllable generative modeling.

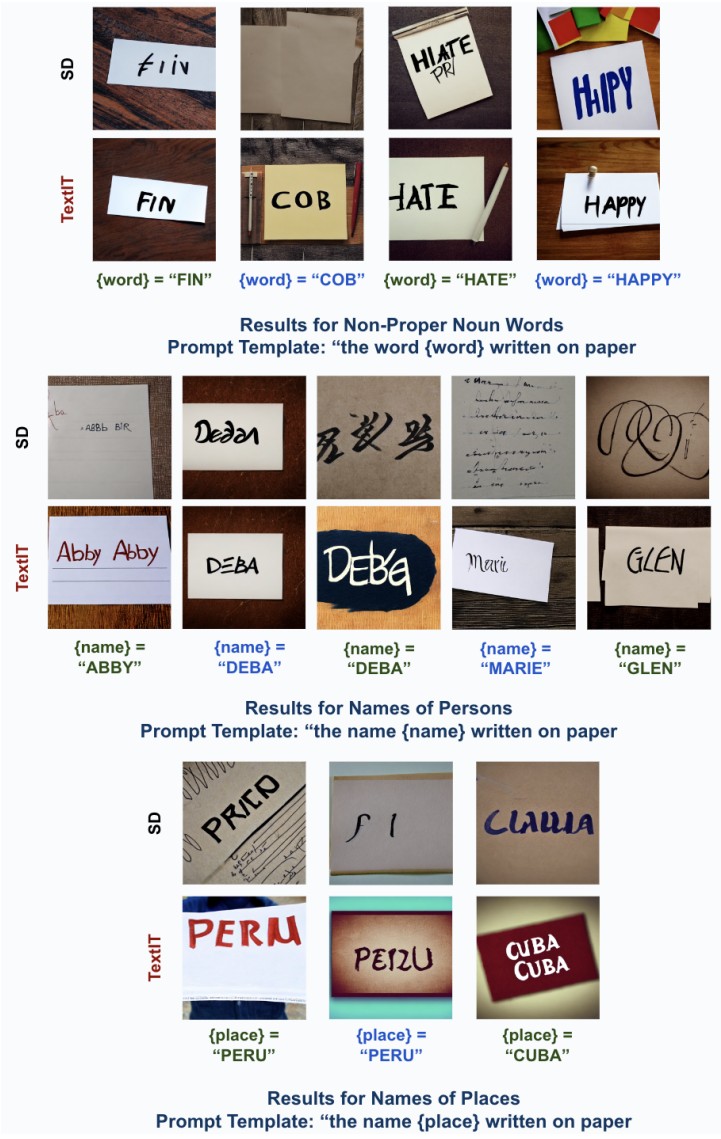

Figure 8: Qualitative Results for Short Text Rendering for various categories: common words, person names, and place names. Certain examples shown here are generated twice using two distinct seeds.

Although here we demonstrate TextIT on single characters and short words to validate the representational alignment and effectiveness of our method, it can generalize to longer text (multiple words and lines) since the optimisation operates sequentially over character-level attention segments. The main bottleneck is computational, not conceptual, and we plan to address this in future work via elegant parallelised implementations of the per-character optimisation.

Moreover, our examples involve simple prompts which mention the target text directly, but our method can be extended to more comprehensive prompts mentioning target words explicitly (an example of which is shown in the Supplementary), as well as use-cases where the target text can be inferred from the prompt through a small language model even when it isn't explicitly mentioned. While this limits applicability to fully open-ended prompts, since TextIT works well given the target text, future work will focus on developing intermediate modules to bridge the gap between prompt and target text, while also ensuring that unintended text and artifacts are suppressed.

Lastly, although the inference-time optimisation, vectorisation, and control points extraction add a few seconds to the inference time compared to the base model, the trade-off for this limitation is that TextIT removes the need for any finetuning, data collection, or computational overhead.

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
