# TextIT: Inference-Time Representation Alignment for Improved Visual Text Generation in Diffusion Models (Supplementary Material)

**Abhikhya Tripathy**
Adobe Research, India
abhikhyat@adobe.com

**Aishwarya Agarwal**
Adobe Research, India
aishagar@adobe.com

**Srikrishna Karanam**
Adobe Research, India
skaranam@adobe.com

**Balaji Vasan Srinivasan**
Adobe Research, India
balsrini@adobe.com

## 1 Appendix

In Section 1.1, we show additional qualitative results for single character visual text generation, which includes results for all 26 letters of the English alphabet. In Section 1.2, we separately compare our method **TextIT** with a recent work called Diff-Text (Zhang et al. [2023a]) because the paradigm in which it operates is close to ours when compared with other concurrent works aiming at improving visual text rendering in diffusion models. In Section 1.3, we provide a result and some analysis around generation of visual text for comprehensive text prompts using **TextIT**. In Section 1.4, we provide a detailed explanation for how the self-attention maps are visualised and how the self-attention loss is calculated. Similarly, in Section 1.6, we provide details about the Focal Loss which we use to calculate our Self-Attention Loss. In Section 1.5, we provide some details around the loss used for rendering small text using our method, and in Section 1.7, we provide additional details about the Control Points that we have used to compute one of our proposed loss components, the Control Points Loss. Finally, in Section 1.8, we end with a discussion on the advantages that our method **TextIT** brings by being an inference time-only method, when compared to other works tackling the problem of visual text generation.

### 1.1 Additional Qualitative Results for Single Character Rendering

In Fig. 1, we present qualitative results for single character visual text rendering, for all 26 capital letters of the English alphabet. Each letter is generated using two distinct Stable Diffusion seeds, allowing for a comparison between our proposed method, **TextIT**, and the baseline Stable Diffusion (SD). The format of prompt used for these results is "the letter {letter}". Our results vividly demonstrate **TextIT**'s efficacy in accurately rendering English alphabet letters while preserving the original image's overall characteristics, thereby maintaining generation diversity. Also note that, in our method, small words are rendered by doing the character-level inference-time optimisation independently for each character in the word, in a loop. Thus, the results in Fig. 1 are also an indicator for the efficacy of our method in the small words generation task.

### 1.2 Comparison of TextIT with Diff-Text

Here, we separately compare our **TextIT** with Diff-Text (Zhang et al. [2023a]) because the paradigm in which it is generating scene text is different from the other methods and is closer to ours. Diff-Text uses as additional control to UNet, Canny edge maps extracted from the target text rendered in a standard font and in black-and-white, similar to how we generate our ground truth attention maps.

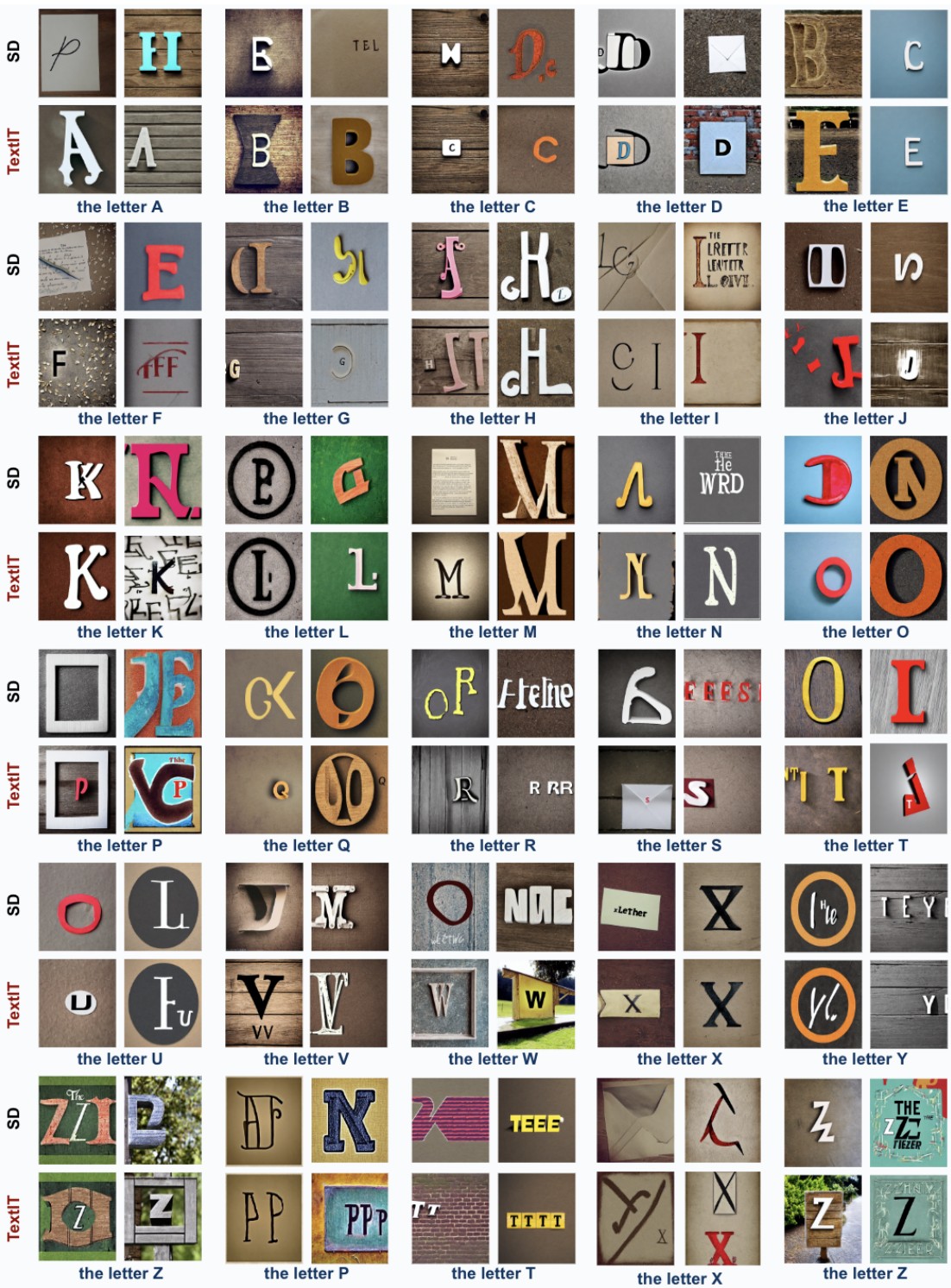

Figure 1: Additional qualitative results for rendering all letters of the English alphabet

Thus, to compare with Diff-Text, we pass our ground truth attention maps for extraction of edges, and use the same input prompt, eg. "the letter A". Qualitative comparison results are shown for letters B, D and A respectively in Fig. 2.

With respect to comparison with Diff-Text as shown in Fig. 2, we see that even though Diff-Text generates correctly rendered single characters, the characters look almost exactly like the ground truth maps (refer Figs. 2 and 4 of main paper to see the font and style), as if the characters have simple been composited on the background with color changes. This shows lack of diversity in their model. In contrast, our method generates results for the same cases with diverse backgrounds, text positions, orientations, fonts and styles.

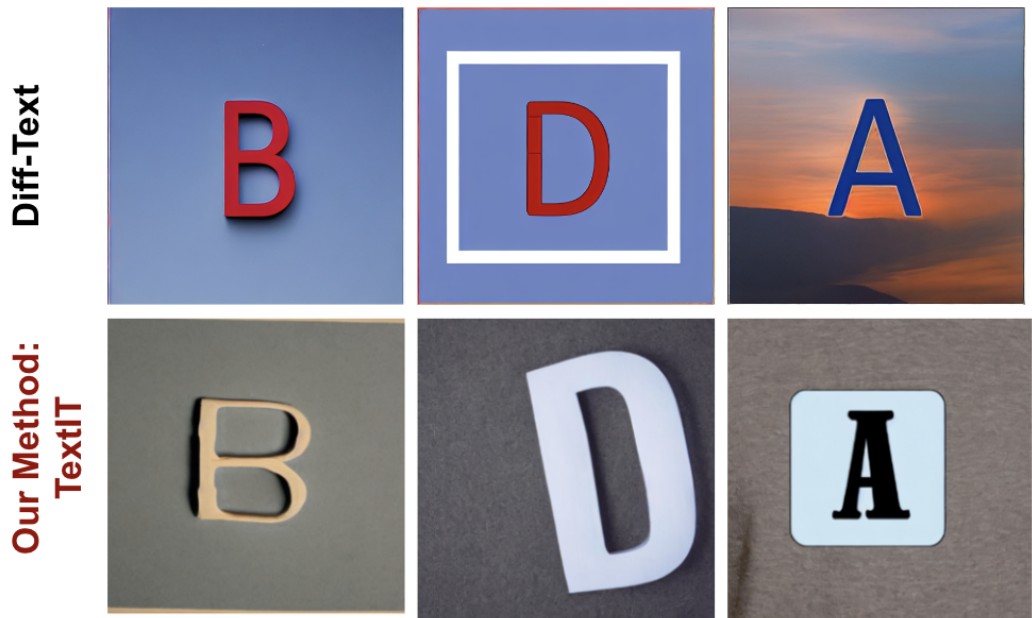

Figure 2: Comparison of **TextIT** results with Diff-Text results

## 1.3 Generation of Visual Text for Comprehensive Prompts

Fig. 3 shows the comparison between generated images from Stable Diffusion and our TextIT for the prompt - **a boy holds a sign saying 'STOP'**. From this we can see that our method is able to perform well even on more comprehensive prompts, where other aspects of the image are described along with the text to be generated. We also see that the output is able to capture the other aspects described in the prompt well.

From Fig. 3, we also see that the overall aspects of the image remains the same as SD output, including the boy's shirt, face and background. This reiterates the fact that **TextIT** outputs fix the rendered text but preserve the overall harmony of output image with respect to the SD generation. This in turn preserves the ability of the base SD model to generate very diverse and creative images.

## 1.4 Details of Self-Attention Loss Computation and Visualisation

Our intuition behind the proposed Self-Attention loss stems from our observations that: **(1)** the Self-Attention maps well represent the layout and shape properties of the generated character, as illustrated in the main paper's Fig. 3, and **(2)** our method evolves the SA maps to more closely resemble the target text glyph, as demonstrated in Fig. 6 of the main paper. Our goal is to guide the self-attention maps to follow the outline of the target character to be rendered. To estimate the same, we compute the Focal Loss between the intermediate self-attention maps and the corresponding Skia-rendered image of the character to be generated, which we use as our ground truth.

At a denoising timestep $t$, we first obtain the self-attention maps computed using $QK^T \in \mathbb{R}^{r \times r \times c}$ (where $r$ is the layer resolution) inside the self-attention block of the UNet. We next perform

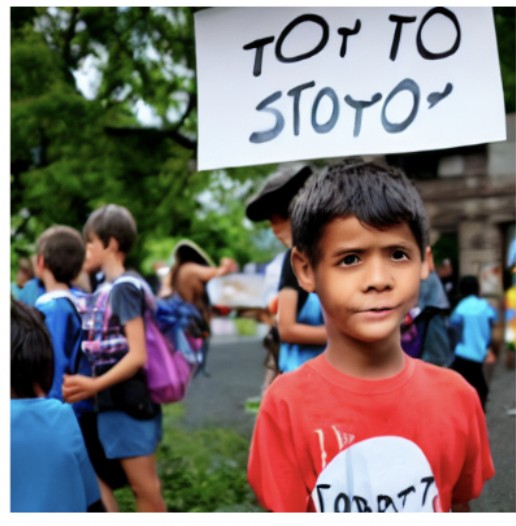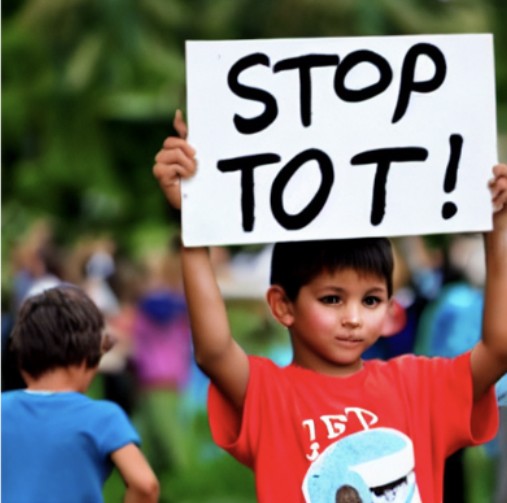

**Stable Diffusion**  **TextIT**

Figure 3: Comprehensive Prompt Result

Principal Component Analysis (PCA) on these maps and retain the first principal component to obtain $A_t \in \mathbb{R}^{r \times r}$. Please note that the self-attention visualizations (in Figs. 3, 4 and 6 of the main paper) also utilise this first principal component $A_t$. Once $A_t$ is computed as described above, we obtain the final ground-truth image $I_{gt}$ after reshaping the corresponding Skia-rendered image as per the layer resolution $r$ and compute the Focal Loss. More details about the calculation of Focal Loss is given in the next section and it is mathematically described in Equation 1.

For computing the Self-Attention loss, we utilise the self-attention layers of resolution $r = 16$, as per our observations of these layers being the most dominant ones in controlling the outline of rendered characters in the generated image. Previous works such as Chefer et al. [2023] have also noted similar observations in the context of layout and shape of natural objects being generated. On the other hand, while computing the control points loss, we utilise the layers of resolution $r = 64$ because the control points could be computed more accurately from the more-detailed higher resolution maps. Note that both the SA maps and the ground truth maps are single-channel gray-scale images.

### 1.5 Details of Loss Used for Rendering Small Text

For generating small text results, we use the Self-Attention only loss optimisation, in order to reduce the time taken for inference-time optimisation, since we are optimising for each character in the target text in a loop (as detailed in the main paper).

### 1.6 Details of Focal Loss

Focal Loss (FL) is a class imbalance mitigation technique, particularly useful when dealing with heavily-skewed datasets. It down-weights well-classified examples and emphasizes hard examples during training. The mathematical expression describing it is:

$$FL(p_t) = -(1 - p_t)^{\gamma} \log(p_t) \tag{1}$$

where $p_t$ is the predicted probability of the true class, and $\gamma$ is a focusing parameter that controls the rate at which easy examples are weighted down.

Thus, FL mitigates class imbalance by penalizing incorrect classifications in proportion to their confidence scores, while reducing the impact of well-classified examples.

## 1.7 Details of Control Points Used

Note that there are two types of control points - Quadratic and Cubic Control Points [1]. Cubic Points follow the glyph shape of the text more closely than Quadratic. This close correspondence might lead to large losses and potential instability. However, keeping this analysis in mind, in our paper we have exclusively used Quadratic Control Points which follow the glyphs more loosely and do not lead to very large losses or instability.

## 1.8 Advantages of an Inference-Time Optimisation-Only Approach

By using a method that operates completely in the inference time, we gain several advantages over existing methods. First, by not requiring any training or fine-tuning, we save significantly on the overhead of computational costs, time and large-scale training data. Collecting large-scale data for text rendering that is suitable for mitigating the generation of gibberish text can be challenging, especially for uncommon words, proper nouns and low-resource languages. All our method needs is a graphics library to render the target text, which can be inferred from the text prompt itself. Our results on small text rendering show how our method improves the visual rendering of such text. Secondly, by not using any additional condition input, we are not changing the base diffusion model (the UNet) at all, thereby retaining all of the creativity of the base model in terms of background diversity and styles and fonts of generated text. This makes our approach advantageous over conditioning-based methods, such as ILVR (Choi et al. [2021]) and ControlNet-based (Zhang et al. [2023b]) approaches. An example of the loss of generation diversity resulting from changing the condition can be seen in the text generated by Diff-Text, as shown in comparison to **TextIT** in Section 1.2. The visual texts generated by Diff-Text are very restricted by the input edge images used for conditioning the UNet. Moreover, our method can be plugged into the inference phase of any backbone diffusion model to ensure improved text rendering.

Our inference-time optimisation routine is only triggered when there is a target text present to be rendered, as inferred from the input prompt, and doesn't alter the base diffusion model through any fine-tuning or additional condition control. Thus, the text-to-image model retains its original performance in terms of diversity and quality of generated images, even when no text is to be generated in the output image. In this case, the output image would be the same as generated by the base Stable Diffusion.