# OpenReview forum: "TextIT: Inference-Time Representation Alignment for Improved Visual Text Generation in Diffusion Models"
_NeurIPS.cc/2025/Workshop/UniReps — UniReps2025_

### Official Review · Reviewer_77ZJ · 2025-09-10
**TextIT: Inference-Time Representation Alignment for Improved Visual Text Generation in Diffusion Models**

**Confidence:** 4

**Review:**

Paper Summary:

This paper introduces TextIT, a novel inference-time optimization framework designed to improve visual text rendering in text-to-image diffusion models without requiring any additional training or fine-tuning. The key innovation lies in manipulating intermediate self-attention (SA) maps during inference to align them with reference maps derived from ground truth text, rendered using standard graphics libraries like Skia. Two types of loss are proposed: (1) Self-Attention Loss (LSA) using Focal Loss on grayscale SA maps, and (2) Control Points Loss (LCP) which compares vectorized Bézier curve control points extracted from SA maps and ground truth renders. The method is implemented on top of Stable Diffusion and shows superior performance in OCR accuracy, CLIP score, and FID when compared to multiple baselines such as TextDiffuser and AnyText.

---

Summary Of Strengths:

The paper proposes a training-free, plug-and-play mechanism that is model-agnostic and works during inference, which is highly practical for real-world deployment. The use of self-attention maps as a control mechanism is insightful, and the control point vectorization based on Bézier curves offers a novel interpretability angle for visual text modeling.

The authors provide comprehensive evaluations, both qualitative and quantitative, including ablation studies that validate the necessity of combining both LSA and LCP losses. They also clearly demonstrate the method’s effectiveness in generating single characters and short words, particularly proper nouns—a known weakness in existing diffusion models. The results show consistent improvement across all metrics (OCR, CLIP, and FID), even when compared against strong baselines, despite their use of extra data or fine-tuning.

The architecture and methodology are clearly explained, and the illustrative figures (e.g., evolution of SA maps, character-level attention) support the claims convincingly. The technique’s generality and modularity (applicable to any text prompt without extra data) are particularly compelling for real-world deployment.

---

Summary Of Weaknesses:

While the method is elegant and promising, it is still restricted to small-scale visual text (single characters and short words). The experiments, though well executed, are limited in scope and do not yet include multi-line or paragraph-level visual text generation, which are critical in real applications (e.g., document generation, posters, infographics).

The paper currently assumes explicit availability of the text to be rendered, which is practical for controlled settings but may limit generalizability to open-ended generation where text is not easily extractable from the prompt.

The selection of baseline models could be expanded to include recent parameter-efficient fine-tuning techniques or prompt-tuning-based methods. The paper briefly mentions omitting works like DreamText and Glyph-ByT5 due to their training-based nature, but a comparison -- even if limited -- would strengthen the empirical rigor.

Another limitation is the lack of user studies or perceptual evaluations, which are increasingly expected for visual generation tasks. While OCR and CLIP are reasonable proxies, a human evaluation of the fidelity and readability of rendered text would be valuable.

Lastly, while the use of Bézier control points is innovative, runtime overhead and the computational cost of extracting and optimizing over control points at multiple timesteps are not discussed in detail.

---

Comments Suggestions And Typos:

* Consider emphasizing in the abstract and introduction that this is the first inference-time-only method for visual text rendering, as that is a key differentiator.
* On page 6 (Lines 230–236), the description of prompts like "The letter A" is a bit simplistic. Consider discussing whether the approach generalizes to more complex prompts like “A red mug with the word ‘Java’ on it.”
* The reliance on Skia for rendering ground truth text is sound, but it would help to mention if the fixed font/style introduces any limitations (e.g., for stylized text).
* Minor typo on L193: “and in black-and-white” -- redundant "and" could be edited for clarity.
* Figures 6–8 could benefit from clearer captions explaining what is shown, especially for non-expert readers.
* Add discussion on latency: how much additional computation time does TextIT introduce over standard inference?
* The authors might want to clarify if this approach can be used in real-time systems or interactive applications.

**Score:**

3

**Topic Fit:**

2

---

### Official Review · Reviewer_2EE1 · 2025-09-14
**Review of Submission 90**

**Confidence:** 4

**Review:**

The paper proposes an improvement on diffusion-based image generation of visual text that does not require additional data or training. During inference the proposed method aligns intermediate self-attention representations of an image being generated with those of a correctly rendered text image. Experimental evidence indicates that the method can outperform existing work both quantitatively and qualitatively, in the context of simple one-character prompts. The supplementary material contains further evidence in support of this conclusion, as well as a demonstration of more comprehensive prompting.

Strengths:
 - The idea of aligning representations to synthetically rendered text, during the diffusion process, is very interesting and seems to strike a good balance between controlling the shape of generated letters and leaving enough stochastic freedom for detail in the final generated image. The model does not seem to overfit to the chosen font of the rendered text.
 - Accomplishing this alignment during the inference process, thus avoiding potentially costly data collection and training, is also very nice.

Weaknesses:
 - The proposed approach requires a separate 2D graphics renderer (like Skia) and vectorizer (like Mang2Vec) during inference to produce ground truth control points. This requirement could potentially limit the practicality of the approach.
 - Most of the paper’s explanations and experiments focus on the generation of single letters on fairly simple backgrounds, for which the proposed method makes a lot of sense. The extension to longer text segments in more intricate scenes receives less attention, and it remains unclear how well one can expect it to perform.
 - The decision to not penalise extra letters or artifacts (in the quantitative evaluation) could be contested, as the paper did set out to generate visual text that is more legible and accurate.

**Score:**

3

**Topic Fit:**

2

---

### Official Review · Reviewer_dZV6 · 2025-09-15
**Review of "TextIT: Inference-Time Representation Alignment for Improved Visual Text Generation in Diffusion Models"**

**Confidence:** 3

**Review:**

Noting the well-known phenomenon of garbled text produced by generative text-to-image diffusion models meant to produce images (eg., Stable Diffusion), the authors propose an inference-time, training-free method they've named TextIt to improve the visual quality of text visible in a rendered image. TextIt optimizes the total loss function in eq. (1) during the denoising process to update latent codes $\mathbf{z}_t$. The total loss is a linear combination of the self-attention loss (direct pixel-intensity loss between the images of the self-attention maps and the ground truth maps) and the control point loss (MSE between the control points from B\'ezier curves of Mang2Vec's SVG output and the ground truth attention maps).

To show TextIt at work, the authors apply their inference-time optimization to Stable Diffusion 1.4 on single-character generation tasks. Experiments are divided into two types:
1. Self-attention loss for the first 5 timesteps followed by control point loss for the following 20.
2. Exclusively self-attention loss for the first 25 timesteps.
The authors note that optimizing beyond 25 timesteps did not seem to produce any marked difference in the intermediate latents. Results are evaluated compared against unaltered Stable Diffusion (their primary baseline) as a model that requires no re-training or finetuning, and against TextDiffuser, TextDiffuser-2, and AnyText which are training-based methods. Character images are evaluated by optical character recognition accuracy, CLIP, and FID. Quantitatve results are presented in Table 1, where the authors highlight TextIt outperforming benchmark methods. This appears to be validated by the visuals in Figure 5.

Overall, the authors report that the control point loss provides fine-grained glyph shape control and that self-attention loss alone aligns attention maps but lacks detailed control, thus that the combined approach yields most accurate text rendering; TextIt successfully renders small words including proper nouns and uncommon words; and that the proposed method is controllable and interpretable.

This reviewer finds the following:

**Strengths**
1. TextIt requires no re-training or finetuning (main strength)
2. TextIt appears to achieve character-level generation control opening up a door to many useful purposes
3. Optimization of the proposed loss function occurs in latent space saving much in computations as compared to a similar scheme in ambient space

**Weaknesses**

For this work to be more impactful, meaningful, and reach a broader audience, especially relatively inexperienced readers, the paper would immensely benefit from a deeper discussion on the intuition behind the why of self-attention in this context (e.g., how SA works on character images, why SA and not another means of information filtering). Likewise for the control points. It would be nice to see the author's hypothesis/intuition on why the application of SA and CP losses in the time duration ratios proposed (5 to 20) works and some ablation studies to explore other duration ratios and compare with the 5-to-20 they describe in the paper.

**Score:**

4

**Topic Fit:**

2